# Long term survival and abnormal liver fat accumulation in mice with specific thymidine kinase 2 deficiency in liver tissue

Qian Zhao[1]©, Xiaoshan Zhou[1]©*, Jingyi Yan[1], Raoul Kuiper[2,3], Sophie Curbo[1], Anna Karlsson[1]

**1** Division of Clinical Microbiology, Department of Laboratory Medicine, Karolinska Institute, Karolinska University Hospital, Stockholm, Sweden, **2** Division of Biomolecular and Cellular Medicine, Department of Laboratory Medicine, Karolinska Institute, Karolinska University Hospital, Stockholm, Sweden, **3** Norwegian Veterinary Institute, Ås, Norway

© These authors contributed equally to this work.
* xiaoshan.zhou@ki.se

**Data Availability Statement:** All relevant data are within the manuscript

**Funding:** A.K 15-0953 to Anna Karlsson, Karolinska Institute, www.ki.se. CAN2016/1342-

## Abstract

Deficiency in thymidine kinase 2 (TK2) causes mitochondrial DNA depletion. Liver mitochondria are severely affected in *Tk2* complete knockout models and have been suggested to play a role in the pathogenesis of the *Tk2* knockout phenotype, characterized by loss of hypodermal fat tissue, growth retardation and reduced life span. Here we report a liver specific *Tk2* knockout (KO) model to further study mechanisms contributing to the phenotypic changes associated with *Tk2* deficiency. Interestingly, the liver specific *Tk2* KO mice had a normal life span despite a much lower mtDNA level in liver tissue. Mitochondrial DNA encoded peptide COXI did not differ between the *Tk2* KO and control mice. However, the relative liver weight was significantly increased in the male *Tk2* KO mouse model. Histology analysis indicated an increased lipid accumulation. We conclude that other enzyme activities can partly compensate *Tk2* deficiency to maintain mtDNA at a low but stable level throughout the life span of the liver specific *Tk2* KO mice. The lower level of mtDNA was sufficient for survival but led to an abnormal lipid accumulation in liver tissue.

## Introduction

Replication of mitochondrial DNA (mtDNA) occurs throughout the cell cycle, independently of the replication of nuclear DNA [1]. The synthesis of deoxyribonucleotides necessary for mtDNA in non-replicating cells is depending on the deoxyribonucleoside kinases thymidine kinase 2 (TK2), deoxyguanosine kinase (DGUOK) and deoxycytidine kinase (dCK) [2]. Two of these enzymes, TK2 and DGUOK, are nuclear encoded but with mitochondrial targeting signals directing these enzymes to the mitochondrial compartment [3]. TK2 phosphorylates deoxythymidine and deoxycytidine to their corresponding monophosphates [3] and DGUOK, in a similar way, phosphorylates deoxyguanosine and deoxyadenosine [2]. TK2 and DGUOK together with nucleoside mono- and diphosphate kinases can provide all four dNTPs for

1345 to Anna Karlsson, The Swedish Cancer Society. https://www.cancerfonden.se/om-oss/about. K2014-66X12162-18-3 to Anna Karlsson. The Swedish Research Council, vr.se/english.html. NO The funders had no role in study design, data collection and analysis, decision to publish, or preparation of the manuscript.

**Competing interests:** The authors have declared that no competing interests exist.

mtDNA synthesis also in non-proliferating cells. Consequently, deficiency in TK2 or DGUOK results in mitochondrial DNA depletion syndromes (MDS) with differences in affected tissue (s) depending on the tissue specific demand on synthesis of mtDNA. Although TK2 and DGUOK catalyze similar reactions and together provide all four dNTPs necessary for mtDNA synthesis, the tissues affected differ in patients with deficiency of any of these enzymes. Humans with TK2 deficiency primarily present with myopathy and neurological symptoms [4] while DGUOK deficiency is associated with liver failure and encephalopathy [5]. We have previously studied a complete Tk2 KO mouse model [6] and it showed impaired beta oxidation function in liver as well as low ketone bodies and glucose in the blood of these mice [7]. We thus hypothesized that altered liver metabolism had a central role for the severe phenotype of the complete Tk2 KO mice. The explanation of differences in tissues affected by TK2 and DGUOK deficiency remains largely unclear although tissues with a high energy demand are generally more severely affected in mitochondrial disorders. Liver is an organ with high levels of mitochondria and thus expected to be sensitive to both TK2 and DGUOK deficiency. To further investigate the mechanism and explore possible treatment strategies, we generated a liver specific Tk2 KO mouse model (livTk2 KO), which was expected to have a longer life span as compared to the complete Tk2 KO mouse model previously studied. To our surprise, the Tk2 liver specific KO mice showed none of the severe symptoms found in the systemic Tk2 KO mice. Although mitochondrial DNA copy number was much lower in the liver specific Tk2 KO mice liver, the mice compensated well for the depletion of mtDNA in the liver during their entire lifetime. However, the lipid metabolism was compromised that indicated an abnormal lipid accumulation.

## Materials and methods

### Ethics statement

The animal experiments were approved by the Linköping's animal ethics committee from the Swedish Board of Agriculture. The ethical permit numbers are: 101–15 and 6487–2021.

### Generation of liver specific TK2 knockout mice

TK2 conditional knockout mice were generated by homologous recombination that replaced a modified TK2 exon V (with additionally two loxp sites, two flp sites, and neo cassette). The TK2 $^{loxP}$/TK2$^{loxP}$ mice were mated to heterozygous albumin cre transgenic mice (+/alb1-cre) to generate double heterozygous mice. Double heterozygous (+/TK2 $^{loxP}$, +/alb1-cre) mice were crossed with TK2 $^{loxP}$/TK2 $^{loxP}$ strain to generate the liver tissue specific knockout mice (TK2$^{-/-}$, +/alb1-cre). The alb1-cre mice were purchased from The Jackson Laboratory. The mice were sacrificed with $CO_2$, and the death was confirmed with cervical dislocation.

### DNA copy number quantification

Total DNA from liver tissue was extracted by using a commercial kit (Qiagen, DNeasy Blood & Tissue Kit 69506). The mtDNA copy numbers were determined by real-time PCR (Applied System 9700). The mouse mt-Nd1 gene and Rpph1 gene were used to measure the mtDNA copy numbers. The standard curve formula was calculated using two plasmids, with each plasmid containing one copy of the mouse gene. The plasmid was diluted to the concentrations designed in advance as standard curve references. According to the standard curve formula, the number of copies for each gene was calculated for each sample, and the number of mtDNA copies per diploid nucleus was calculated according to the formula: mtDNA copies per diploid nucleus = 2 × (mt-Nd1 gene copies/Rpph1 gene copies) [8].

## Primers

The mouse mt-Nd1 gene (mitochondrial DNA encoded NADH dehydrogenase 1; forward primer: 5′-TCGACCTGACAGAAGGAGAATCA, reverse primer: 5′-GGGCCGGCTGCGTATT, probe: FAM-AATTAGTATCAGGGTTTAACG-TAMRA).

The mouse Rpph1 gene (nuclear DNA encoded ribonuclease P RNA component H1; forward primer: 5′-GGAGAGTAGTCTGAATTGGGTTATGAG, reverse primer: 5′-CAGCAGTG CGAGTTCAATGG, probe: FAM-CCGGGAGGTGCCTC-TAMRA).

The screening primers for alb-cre mice forward primer: 5′-CCAATTTACTGACCGTACACC, reverse primer: 5′-GTTTCACTATCCAGGTTACGG. The screening primers for conditional knockout mice forward primer: 5′-TCTCCTCTTCCTCAT CTCC, reverse primer: 5′-GTGGTCGAATGGGCAGGT.

## Gene expression determination with real-time qPCR

Total RNA was extracted from mouse livers with RNeasy kit (QIAGENQ, 74106, Sofielundsvä-gen 4, Sollentuna, Sweden) and cDNA was synthesized with High-Capacity cDNA Reverse Transcription Kit (Applied Biosystems, Life Technologies Corporation|Carlsbad, CA, 4368814). Real-time qPCR was performed using KAPA SYBR® Fast qPCR Master Mix (2X) Universal (Kapa Biosystems, Merck KGAA, Darmstadt, Germany, KM4602) in ABI 7500 Fast system (Applied Biosystems). The sequences of primers are listed in the S1 Table. The house-keeping gene *Gapdh* was used as loading control for the analyses.

## Western blot analysis and histology

At 1.5 years of age, the mice were sacrificed, and organs were saved for histology. The organs were fixed in 4% formaldehyde, snap- frozen in liquid nitrogen, and kept at - 80˚C. For the histology, each group of control and knockout included three samples. Western blot analysis was performed following our group's previous study [8]. The protein concentrations were deter-mined by using the Bradford assay (Quick Start™ Bradford Protein Assay Kit 1, REF: 5000201). Antibodies include mouse monoclonal to Succinate Dehydrogenase Complex Flavoprotein Subunit A (SDHA) (Abcam Inc, Cambridge, MA, USA, ab14715), mouse monoclonal to Mito-chondrially Encoded Cytochrome C Oxidase I(MTCO1) (Abcam Inc, Cambridge, MA, USA, ab14705), anti-voltage-dependent anion-selective channel (VDAC) antibody (Santa Cruz Bio-technology, Inc. Dallas, Texas, U.S.A. sc390996), Dako Polyclonal Rabbit anti-Mouse Immu-noglobulins/HRP (Santa Clara, CA, United States, P0260).

## Statistics

Body weights, organ weights, relative organ weights and mtDNA copy numbers were analyzed with student's t-test (unpaired two-tailed). The Mann-Whitney test was used to compare the control to the knockout groups in body weight data analysis. Significant level was set to $p < 0.05$.

## Results

### Generation and characterization of liver specific Tk2 KO mice

The livTk2 KO mice were generated based on the Tk2 conditional KO mice (Fig 1A) and transgenic mice (+/alb1-cre). The mice with the albumin cre transgenic gene were crossed with Tk2 conditional KO mice, and thus the offspring carried both alb1-cre and the modified Tk2 gene. Those double heterozygous mice were intercrossed, constructing an offspring with both the homozygous $Tk2^{loxP/loxP}$ and the alb1-cre transgenic gene. Subsequently, the albumin

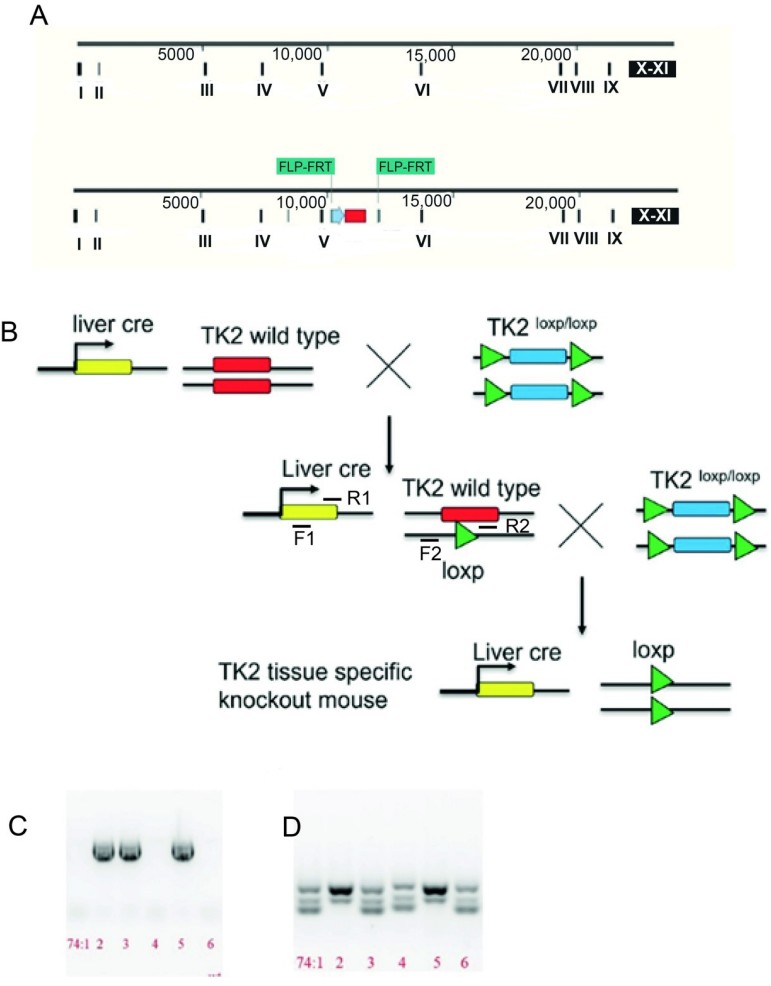

**Fig 1. Generation of liver specific Tk2 knockout mouse model.** (A) Generation of targeting construct for deleting exon V in TK2 through homologous recombination. (B) Breeding strategy of generating livTk2 KO mice. (C and D) Screening of livTk2 KO mice through combining Alb-cre-specific primers (F1 and R1) and mutated-allele-specific primers (F2 and R2). Lanes 2 and 5 indicate homozygous livTk KO mice.

cre enzyme deleted the Tk2 exon V specifically in hepatocytes (Fig 1B). The mice were born apparently normal. After 3 weeks, the mice were weaned. The punctured ear tissues were used for genotyping. The alb1-cre gene and Tk2 allele with loxp sites were amplified with specific primers. The result showed mice positive with both alb1-cre (Fig 1C) and Tk2 conditional primers (Fig 1D). The mice were observed, and body weights were recorded for 55 weeks, for both control and livTk2 KO mice. The weights of the mice were grouped into male (Fig 2A) and female mice (Fig 2B). Both control and livTk2 KO mice body weights were increased during the time observed. In both male and female mice, the average weights of controls were slightly higher than in the KO mice from about 12 weeks until 55 weeks, although the differences were not significant.

## Long term effects in the livTk2 KO mice

The mice were sacrificed at the age of 1.5-years-old and analysis was performed. For the male mice, the average body weight of the livTk2 KO mice was not significantly changed compared to that in control mice (Fig 3A). Moreover, the absolute weights of liver, kidney, and heart were

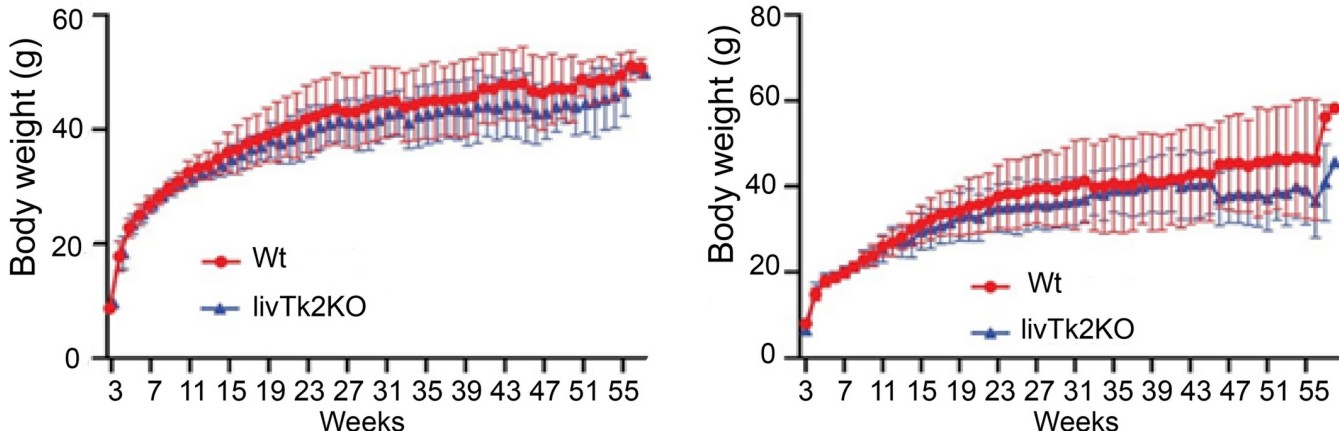

**Fig 2. Body weights.** (A) Male mice and (B) female mice body weights were measured from week 3 to week 55. 20 mice in control group (n = 10 for each gender) and 8 mice in knockout group (n = 4 for each gender). The error bars indicate standard deviation. The Mann-Whitney test was used to compare the control to the knockout groups. Significant levels were set to p < 0.05(*).

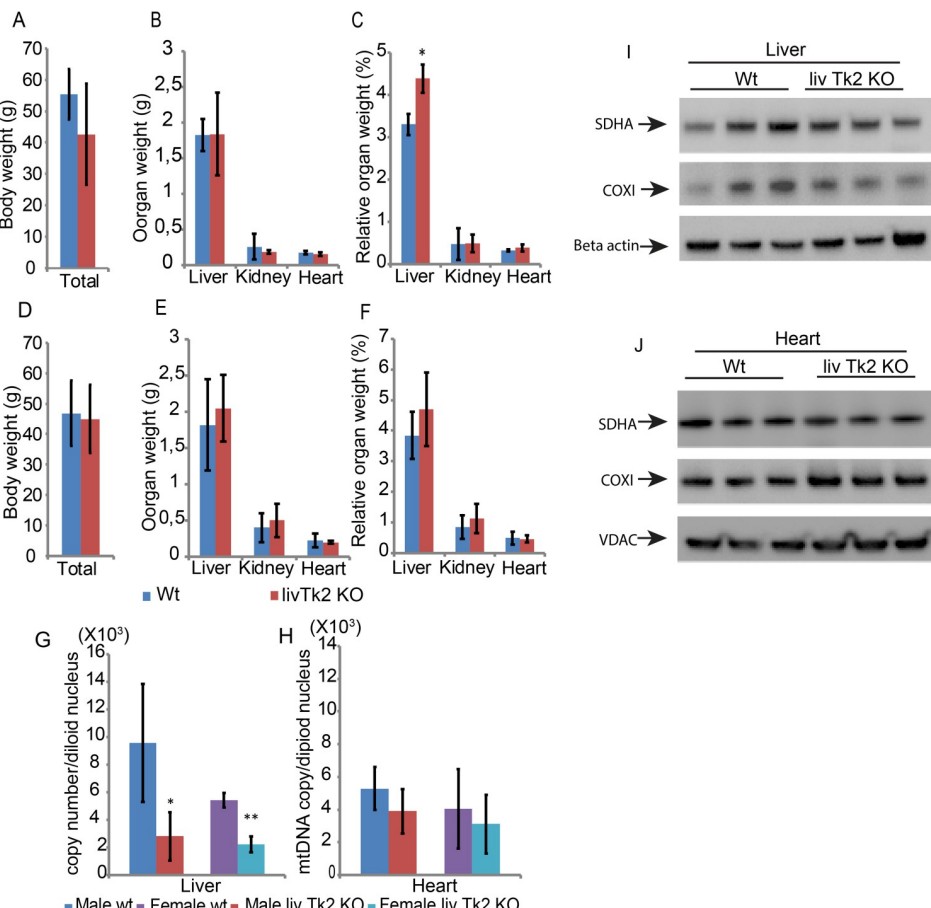

**Fig 3. 1.5-year-old mice body weight and mtDNA copy number.** Male mice body weights (A), organ weights (B), and relative organ weights (C). Female mice body weights (D), organ weights (E), and relative organ weights (F). Mitochondrial DNA copy number from liver (G) and heart (H). SDHA and COX I protein levels in liver (I) and heart (J). n = 4 for each group, and error bars indicate standard deviation. The Mann-Whitney test was used to compare the control to the knockout groups. Significant levels were set to p < 0.05(*).

similar between the control and livTk2 KO groups (Fig 3B). The relative organ weights (absolute organ weight vs total body weight), however, were different for liver but not for kidney or heart. The livTk2 KO male mice had a higher relative liver weight than that in the control mice (Fig 3C). In female mice, the body weights were similar in the control and livTk2 KO groups (Fig 3D). Moreover, the relative organ weights in livTk2 KO mice were not changed compared to the controls (Fig 3E and 3F). The liver tissue mtDNA copy number was significantly lower in both male and female livTk2 KO mice as compared to the control group (Fig 3G). Male livTk2 KO mice had 30% of mtDNA copy number compared to the control group of male mice (Fig 3G). Female livTk2 KO group mice had about 40% of mtDNA copy number compared to the control group of female mice (Fig 3G). The heart mtDNA copy numbers from male and female mice were not significantly different between the control and KO groups (Fig 3H). The investigated mitochondrial proteins, COX1 and SDH, showed similar levels in the livTk2 KO mice and the control mice, in both liver and heart tissue (Fig 3I and 3J).

### Histopathology and electron microscopy of the livTk2 KO mice

Histopathology revealed some minor alterations in the liver of both control and KO mice. This was shown as diffuse mild micro vesicular steatosis, occasional pigmented Kupffer cells, and mild portal fibrosis, as observed in both groups of 1.5 years old mice (Fig 4A and 4B). In the livTk2 KO mice, mild focal macrovascular lipidosis and areas of cells with clearer cytoplasm were observed. In addition, the livTk2 KO mice showed frequent foci of granulomatous inflammation and pigmented macrophages. The inflammation included terminal hepatic veins, endothelial involvement and multifocal necrosis associated with inflammation and occasional microhemorrhage (Fig 4B). We did not observe clear morphological differences in either heart (Fig 4C and 4D) nor liver (Fig 4E and 4F) using electron microscopy.

### Decreased fatty acid synthesis in the livers in livTk2 KO mice

To understand the mechanisms underlying the mtDNA depletion and abnormal lipid metabolism, we measured expression of genes involved in mtDNA maintenance and lipid metabolisms. As shown in Fig 5, the liver *Tk2* level in livTK2 KO mice was lower than that in wild type. However, the difference was not significant. The expression levels of other enzymes in both nucleoside salvage pathway (including *dCK*) and de novo pathway (including ribonucleotide reductases: *Rmr1*, *Rmr2* and *Rmr2b*) were not altered. Moreover, *Tfam* and *Ppargc1a*, involved in mtDNA maintenance and mitochondrial biogenesis, respectively, were also not changed. Interestingly, we found that fatty acid synthase (*Fasn*) was significantly lower in livTk2KO mice than that in wild type mice. Consistent with down-regulated *Fasn*, the expression level of sterol regulatory element-binding transcription factor 1 (*Serbp1*) was also low in livTk2KO mice compared to wild type mice. The expression level of *Cpt1*, responsible for transport fatty acids across mitochondrial membrane, was not changed in livTk2KO mice.

### Discussion

Total Tk2 knockout mice survived for around two weeks with a rapid decline of mtDNA levels in investigated tissues, starting at day seven [6]. These mice were born normal and thus evidently had the ability to synthesize mtDNA during the fetal stage. However, Tk2 deficiency resulted in a rapid decline of mtDNA levels after birth, initially in skeletal muscle and at about two weeks of age also in liver. The liver tissue of total Tk2 KO mice had impaired beta oxidation and altered fatty acid metabolism [6]. Since our hypothesis was that liver tissue had a central role in the phenotypic alterations and short survival of the total Tk2 KO mouse model, we constructed a liver specific Tk2 KO mouse model to further investigate metabolic alterations

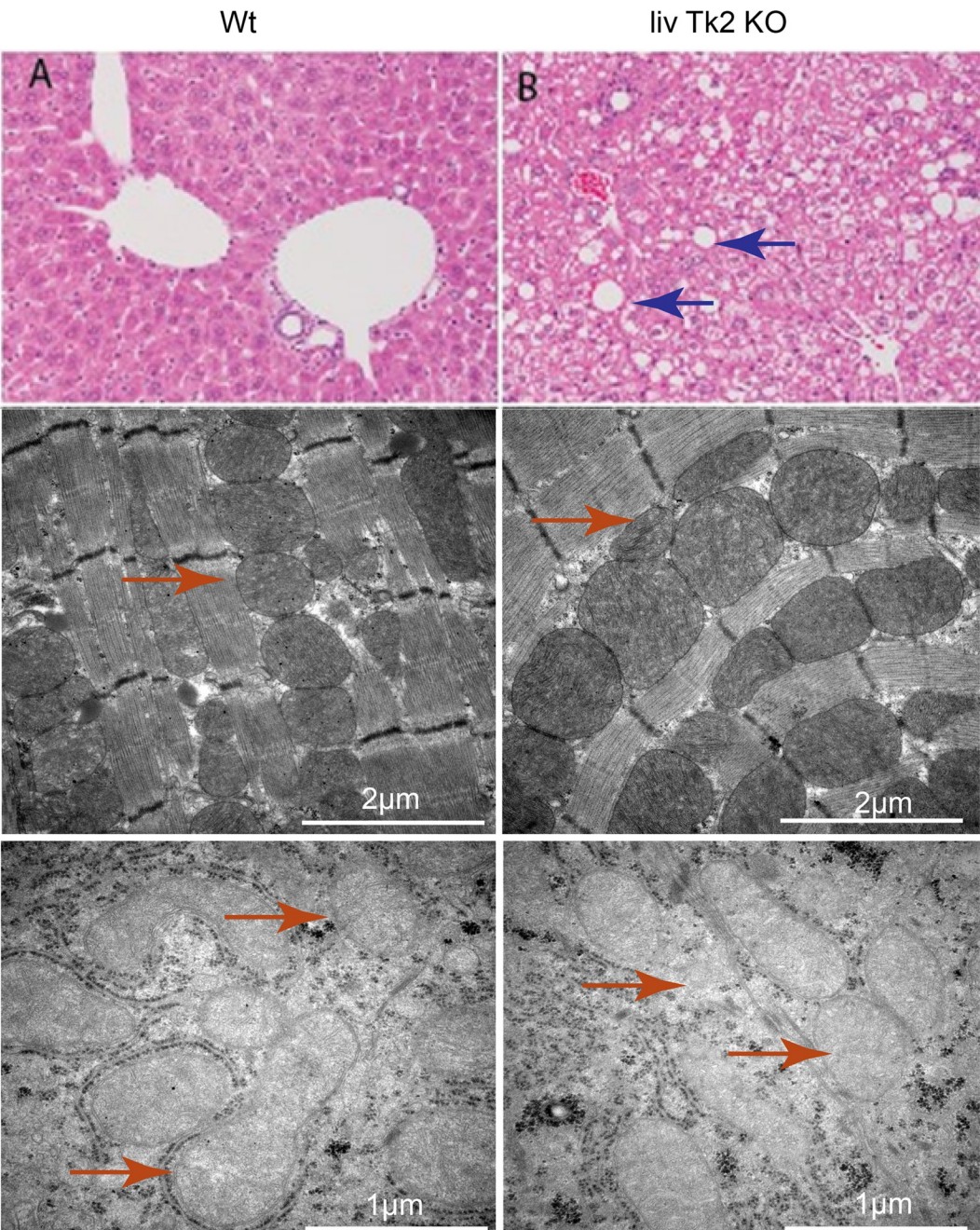

**Fig 4. Histopathology and electron microscopy results.** Histopathology analysis of liver tissues from control (A) and livTk2 KO mice (B). Electron microscopy analysis of heart tissues in control (C) and in livTk2 KO mice (D). Electron microscopy analysis of liver tissues in control (E) and livTk2 KO mice (F). Blue arrows indicate lipid droplets in liver and red arrows indicate mitochondria in heart and liver tissues.

as a result of mtDNA depletion in liver. Surprisingly, the mice lacking Tk2 in the liver were growing and behaving apparently normal with around 30% of mtDNA level in the livers compared to wild type mice. DGUOK provides dAMP and dGMP for mtDNA replication. It has been shown that knockout of *Dguok* could lead to 95% reduction in mtDNA copy number in livers [9]. Therefore, our observation indicates liver has higher compensatory capacity to the

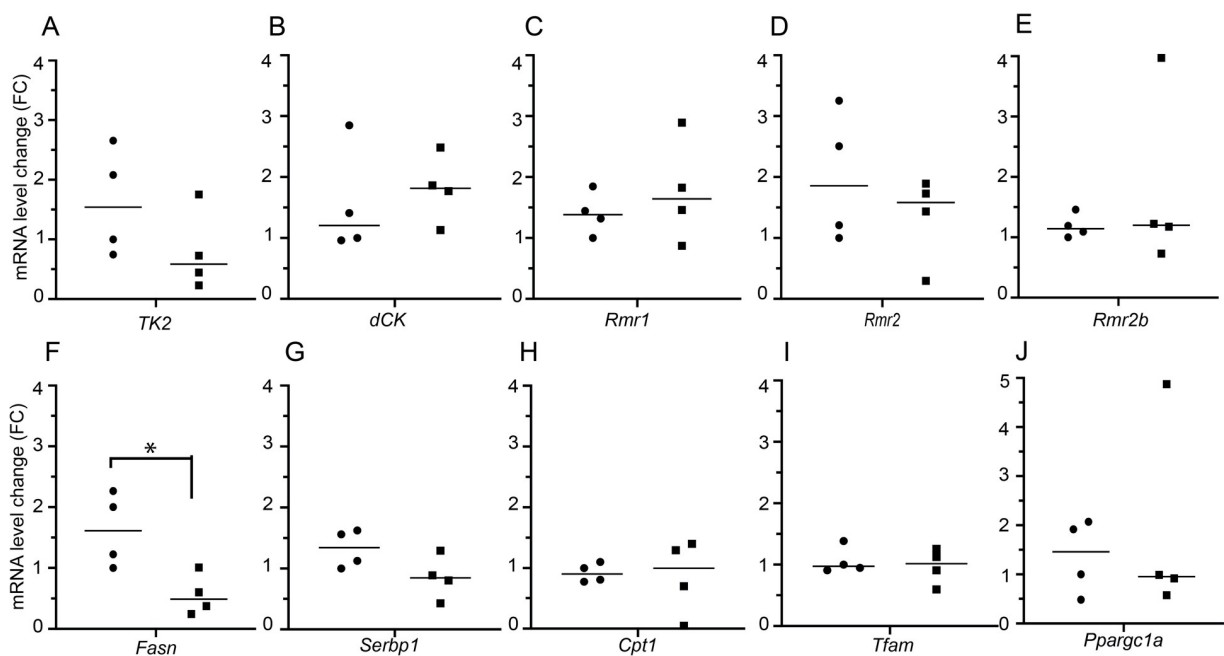

**Fig 5. Expression levels of genes involved in mtDNA maintenance and lipid metabolism.** (A-J) *Tk2*, *dCKs*, *Rmr1*, *Rmr2*, *Rmr2b*, *Fasn*, *Serbp1*, *Cpt1gene*, *Tfam* and *Ppargc1α*, respectively. Data presented as mean ± SD. Statistically significant difference (two-tailed unpaired Student's t-test), *p<0.05. FC: Fold change (livTk2 KO vs wild type).

deficiency in TK2 activity than in DGUOK activity. The substrates of TK2, dThd and dCyd, are known to be small, soluble and can pass through membranes upon concentrative nucleoside transporters (CNT) and equilibrative nucleoside transporters (ENT) [10–13]. The main treatment of patients with TK2 deficiency is by supplementation of deoxythymidine (dT), the substrate for TK2 [14]. Initial studies were performed using supplementation with dTMP, the product catalyzed by TK2 [15]. However, comparison of the addition of dT and dTMP did not show significant differences and may probably be explained by the low stability of dTMP that is rapidly hydrolyzed to dT [15, 16]. The model presented here could be used to study the contribution of labelled dT to mtDNA synthesis in liver tissue.

Our total Tk2 knockout mice only survived approximately around two weeks with mtDNA depletion in multiple organs, especially in skeletal muscle and brain. In their liver tissues mtDNA copy number rapidly declined to approximately 20% between day 7 and 14, as compared to control mice [6]. In the present study, we found that the liver mtDNA levels were 30% and 40% in 1.5-years-old livTk2 KO male and female mice, respectively, which are higher than that in total Tk2 knockout mice. It has been demonstrated that the albumin Cre transgene is effective in fetal and neonatal mice [17]. Therefore, the TK2 activity in the livers in our livTk2 KO mice should be disrupted at birth. Since both total TK2 KO and livTk2 KO mice were born normal, the difference in mtDNA copy number suggests that maintenance of mtDNA level in liver tissue involves other mechanisms. Since thymidine kinase 1 (TK1) is cell cycle dependent, and dCK is constitutively expressed throughout the cell cycle and provides all four dNTPs for mtDNA replication [18, 19], the compensation of the loss of TK2 in the livTk2 KO mice was probably from dCK activity. However, we did not observe up-regulated mRNA levels of *dCK*, *Rmr1*, *Rmr2 and Rmr2b* in livTk2 Ko mice. The partly compensation of dTTP and dCTP might result from increased activities of these compensatory enzymes, but not from increased gene expression. Accumulating evidence indicates that whole mitochondria

including their mtDNA can be transferred intercellularly [20]. It has been demonstrated that such horizontal mtDNA transfer was a common phenomenon during mouse development and probably acted as a compensatory mechanism to improve mitochondrial function [21]. However, whether horizontal mitochondria transfer played a role in the maintenance of mtDNA copy number in the livTk2 KO needs further study. We also tried to confirm the knockout of *Tk2* in liver with real-time qPCR. However, it was difficult to design a pair of optimal primers since exon 5 is short. Therefore, the detected TK2 mRNAs in the qPCR analysis might be from an unspecific PCR amplification.

The only observed difference between the liver specific Tk2 KO mice compared to controls was a slight decrease of body weights over the 1.5 years observation period and a significantly increased relative liver weight in male livTk2 KO mice. The similar tendency of increasing relative liver weight in female mice was also observed. These results indicated that the lower mtDNA level was sufficient for most of the function of liver and mouse survival. Electron microscopy analysis also confirmed the normal morphology of cells and normal mitochondria in liver and heart. However, the lipid metabolism was compromised with increased lipid accumulation. Considering the lower levels of *Fasn* and *Serbp1* and the decreased body weight in livTk2Ko mice, we speculated that the fatty acids synthesis was downregulated in the livers of livTk2KO mice, the liver accumulated lipid was from organs outside liver. mtDNA deficiency would reduce the polypeptide of respiratory complex, decrease the ATP production from the electron transport chain, and thus cause an energetic decline. The COX I and SDHA protein levels did not change, strongly suggesting that the mitochondrial complexes and mitochondrial respiration were not affected by the 70% (male) and 60% (female) decline in mtDNA copy number in liver tissues. Therefore, the altered lipid metabolism might result from other mechanisms.

In conclusion, the liver specific Tk2 knockout model showed that life expectancy was not affected although the level of mtDNA was decreased to 30% of normal mtDNA in hepatocytes of male mice, at least not when the mice were kept at un-stressful conditions. Future studies should address questions regarding what compensatory mechanisms enable a low, but stable, mtDNA level in Tk2 deficient liver tissue. To understand this may be useful to develop treatment strategies in Tk2 deficiency and possibly other conditions affecting mtDNA.

## Supporting information

**S1 Table. Primer sequences for real-time PCR analysis.**
(DOCX)

**S1 Raw images.**
(PDF)

## Author Contributions

**Data curation:** Qian Zhao, Xiaoshan Zhou, Jingyi Yan, Raoul Kuiper.

**Formal analysis:** Sophie Curbo.

**Funding acquisition:** Anna Karlsson.

**Project administration:** Anna Karlsson.

**Supervision:** Xiaoshan Zhou, Sophie Curbo, Anna Karlsson.

**Writing – original draft:** Anna Karlsson.

**Writing – review & editing:** Xiaoshan Zhou, Raoul Kuiper, Sophie Curbo, Anna Karlsson.

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
