## [Decision Letter · Decision Letter 0]

16 May 2023

PONE-D-23-11585Long term survival but abnormal fat accumulation in mice with specific thymidine kinase 2 deficiency in liver tissuePLOS ONE

Dear Dr. Zhou,

Thank you for submitting your manuscript to PLOS ONE. After careful consideration, we feel that it has merit but does not fully meet PLOS ONE’s publication criteria as it currently stands. Therefore, we invite you to submit a revised version of the manuscript that addresses the points raised during the review process.

We look forward to receiving your revised manuscript.

Kind regards,

David C. Samuels

Academic Editor

PLOS ONE

Journal Requirements:

Additional Editor Comments:

Both reviewers point out significant scientific details that are missing in your report. Please review their comments carefully respond fully in your revision.

Reviewers' comments:

Reviewer's Responses to Questions

**Comments to the Author**

1. Is the manuscript technically sound, and do the data support the conclusions?

Reviewer #1: Yes

Reviewer #2: Partly

2. Has the statistical analysis been performed appropriately and rigorously? 

Reviewer #1: Yes

Reviewer #2: I Don't Know

3. Have the authors made all data underlying the findings in their manuscript fully available?

Reviewer #1: Yes

Reviewer #2: Yes

4. Is the manuscript presented in an intelligible fashion and written in standard English?

Reviewer #1: No

Reviewer #2: Yes

5. Review Comments to the Author

Reviewer #1: Manuscript by Zhao et al., described a liver specific TK2 knockout mouse model created in their lab. The liver specific TK2 KO mice had a normal lifespan even though the level of liver mtDNA was significantly reduced (60% reduction in female and 70% reduction in male) in 1.5 years old mice. They also found lipid accumulation and other abnormalities in liver tissue but other organs were appear normal.

It is not strange to me that the liver specific TK2 knockout mice appear normal and have a normal lifespan since the liver mtDNA levels in 1.5 years old knockout mice, although significantly reduced, are still significantly higher than the threshold levels (< 20%) that cause mitochondrial malfunction or diseases. Their results indicated that liver has a compensatory mechanism for dTTP and dCTP synthesis that can partly supply the dTTP and dCTP needed for liver mtDNA synthesis but they did not provide any experimental data for it. I think this possible compensatory mechanism should be explored or at least discussed in detail in the manuscript. The authors proposed also that transport of dNMP between cells or organs may be another compensatory mechanism but they did not provide any experimental data support or reference. I think these are important issues and should be addressed.

Minor points.

1. Should specify abnormal fat accumulation occurs in lever tissue in the title.

2. Page 3 line 49: “directing their enzyme activity to the mitochondrial compartment” should be “directing these enzymes to the mitochondrial compartment”.

3. Page 3 line 50: “mono-phosphate” should be “monophosphate”

4. Page 3 line 51 and 52: should be TK2 and DGUOK together with nucleoside mono- and diphosphate kinase can provide all four dNTPs since TK2 and DGUOK can only provide dNMPs.

5. Figure legends should provide more information to make them easier to understand!

6. Page 8 line 164, the authors should clarify “relative organ weights”, relative to what?

7. Page 8 line 167, what does the “net weights” mean? Is it total body weight? If so, “body weight” should be used instead.

8. Page 10 line 215, “the products of TK2, dThd and dCyd, are”. dThd and dCyd are substrates of TK2 therefore this sentence need to rephrase.

9. Page 10 line 218 and 219, “substitution” should be “supplementation”.

10. Page 10 line 226, “catabolism” should be “metabolism” since converting ribonucleotides to deoxyribonucleotides is an anabolic process”.

11. Page 10 line 227 and 228, please provide experimental data/evidence or reference for how cells and organs share or transport nucleoside monophosphates.

12. Page 10 line 232, “it seems like that the level kept is low but stable through the life span”. What are the mtDNA levels at earlier stage of life? Have the authors measured mtDNA levels at difference ages to support this statement?

Reviewer #2: The authors describe a new tissue-specific knockout for mitochondrial thymidine kinase 2 (TK2) in the liver. Whole-body knockout mice for the same gene manifest severe encephalomyopathy and die shortly after birth. Previous works, by the same team, on these whole-body knockouts evidenced an important deregulation of hepatic lipid metabolism which could be relevant to the overall disease phenotype. Specific ablation of TK2 in the liver may help elucidate the contribution of liver dysfunction to TK2 deficiency. Interestingly, the authors show how the liver-specific TK2-KO mice survive up to 1.5 years in spite of displaying a severe loss of mtDNA in the liver.

General comments:

I wish the authors could contextualize and compare the results presented in this manuscript metabolism to their previous data with the whole-body knockout. Could the authors add any more data on lipid metabolism in the reported mice that they can relate to the observed in the whole-body knockouts? Did the whole-body knockout present with a similar mtDNA depletion in the liver? The authors propose compensatory mechanisms, perhaps fuelling dNTP synthesis by TK2-independent pathways, even if from other tissues, that would explain the stabilization of mtDNA levels in liver-specific knockout mice. It would be very interesting to provide any data supporting that hypothesis. Have the authors measured deoxynucleoside levels in the liver or plasma?

I think the authors should consider cytosolic phosphorylation of dCtd and dThd by deoxycytidine kinase (dCK) as a likely contributor to maintaining liver mtDNA levels in the tissue-specific knockouts.

On the other hand, I miss the author’s discussion on other mouse models with a similar or even bigger degree of mtDNA depletion in the liver, some generated in their lab (i.e Dguok knockout mice) who have not shortened survival in spite of also manifesting with major alterations in lipid metabolism. Could demands on liver mitochondrial function or mtDNA copy number be different between human and mice?

The authors claim that ‘These mice were born normal and thus evidently had the ability to synthesize mtDNA during the fetal stage’. I wonder if this could have anything to do with the profile of the liver-specific cre-recombinase expression. Is it active from birth? Before? Have the authors checked for mtDNA copy number at any earlier timepoint?

It is not clear why the authors choose to perform most of the analyses in liver and heart. Have they checked for the cre-recombinase efficiency in the liver? Perhaps by assessing TK2 protein or Tk2 RNA levels in the liver? Is it possible residual mtDNA (30%) comes at least partly, from not knocked-out hepatocytes/liver cells?

Have they check for the cre-recombinase specificity? Perhaps checking TK2 protein/RNA expression in tissues other than liver, i.e heart and kidney?

Overall, I find the authors provide scarce data on how marked mtDNA depletion affects mitochondrial function in the liver-specific TK2 knockouts which make it difficult to reach further conclusions beyond the fact that in mice, low mtDNA levels seem to be compatible with life.

Also, I recommend a review of English usage throughout the text. There are also some redundant comments/sentences in the text that could be reviewed.

Specific comments:

- In order to determine whether the statistical value has been correctly assessed sample size should be included for all the experiments. Also please specify whether the error bars indicate an error or standard deviation or better show individual samples distribution (for instance with a dot blot type of graph)

- The authors mention a very limited number of knockout mice included in the weight monitoring graph. Could they explain why the number is so little if survival is not reduced?

- I find some observations to not be fully supported by the data:

* ‘The body weights of the male control mice were higher on average than in the livTk2 KO mice’. I understand from the graph that such a difference is not statistically validated.

*‘The investigated mitochondrial proteins, COX1 and SDH, showed similar levels in the livTk2 KO mice and the control mice, in both liver and heart tissue’. The authors show no quantification of the mentioned proteins but based on the included images, one can appreciate some differences between genotypes that are not discussed by the authors (i.e. increased heart COXI in the KO).

Also, it could be advisable to check for the level of expression of additional respiratory chain subunits, even if nuclear-encoded, since steady-state levels will reflect synthesis but also stability and this may be variable between proteins. BN-PAGES or activity assays on full respiratory complexes would help generate a broader picture of the molecular phenotype in these mice.

- Males are known to weigh more than age-matched females: ‘The average of male mice body weights was higher than the female mice, when comparing control group male mice to control female mice, and livTk2 KO male mice to livTk2 KO female mice’. Thus, I suggest the authors reformulate this sentence (i.e. As expected, ….).

- Figure 1.

Could the authors locate in the diagram the position of the primers used in the genotyping in Fig 1D?

- Figure 4:

Can the images resolution be improved?

What do the scale bars indicate in units? Comparison of Wt and KO images should be performed on images captured at an equivalent scale preferably (I am not sure this is the case).

Also, the heart images may not be representative for the author’s conclusion, since they claim to observe no differences between WT and KO mice while the included images could suggest a different morphology for mitochondria in both genotypes). Could the authors comment on that?

It could be helpful to add some arrows indicating the described features, especially in the HE stainings.

- Line 204 ‘Total Tk2 knockout mice survived for three weeks’ and line 238 ‘Since the total Tk2 KO mice only survived approximately three weeks’.

It could perhaps be more accurate to talk about average survival and not maximal survival. I think on average, whole-body Tk2 knockout mice live around 16 days.

- I think in Line 209 the authors refer to cite number 7 in the reference list

6. PLOS authors have the option to publish the peer review history of their article (what does this mean?). If published, this will include your full peer review and any attached files.

Reviewer #1: No

Reviewer #2: No

---

## [Author Response · Author response to Decision Letter 0]

10 Jul 2023

Dear reviewers and editor,

We have made the changes according to the journal requirments and the comments from the two reiviewers. Below are the response to the comments from the reviewers.

For the reviewer 1:

The major points from the reviewer 1: It is not strange to me that the liver specific TK2 knockout mice appear normal and have a normal lifespan since the liver mtDNA levels in 1.5 years old knockout mice, although significantly reduced, are still significantly higher than the threshold levels (< 20%) that cause mitochondrial malfunction or diseases. Their results indicated that liver has a compensatory mechanism for dTTP and dCTP synthesis that can partly supply the dTTP and dCTP needed for liver mtDNA synthesis, but they did not provide any experimental data for it. I think this possible compensatory mechanism should be explored or at least discussed in detail in the manuscript. The authors proposed also that transport of dNMP between cells or organs may be another compensatory mechanism, but they did not provide any experimental data support or reference. I think these are important issues and should be addressed.

Response: To answer the questions, we performed new experiments to analyze the mRNA level of relevant enzymes providing dNTPs for mtDNA replication both in salvage pathway (including dCK) and in de novo pathway (including Rmr1, Rmr2 and Rmr2b). We did not observe significantly changes in the mRNA levels of these enzymes between wild and livTk2KO mice. The partly compensation of dTTP and dCTP might result from increased activities of these compensatory enzymes, but not from increased gene expression. However, the underlying mechanisms needs to be further explored in future study. Moreover, we also hypothesized that intercellular mitochondria horizontal transfer might also be a mechanism since liver tissue has 40% mesenchymal cells, for instance stellate and Kupffer cells. Our albumin cre mouse can only disrupt Tk2 in hepatocytes. Therefore, the mesenchymal cells still have intact TK2 activity. Additionally, we also measured genes involved in lipid metabolism (Cpt1, Fasn and Serbp1), we found that fatty acid synthesis was significantly downregulated, which indicates an altered lipid metabolism in livTk2KO mice. We have added the results into the manuscript and interpreted the results in the dissection part.

The minor points 

1. Should specify abnormal fat accumulation occurs in lever tissue in the title.

Response: We have changed the title from “Long term survival but abnormal fat accumulation in mice with specific thymidine kinase 2 deficiency in liver tissue” into “Long term survival and abnormal liver fat accumulation in mice with specific thymidine kinase 2 deficiency in liver tissue’’.

2. Page 3 line 49: “directing their enzyme activity to the mitochondrial compartment” should be “directing these enzymes to the mitochondrial compartment”.

Response: We have changed it accordingly.

3. Page 3 line 50: “mono-phosphate” should be “monophosphate”

Response: We have changed it accordingly.

4. Page 3 line 51 and 52: should be TK2 and DGUOK together with nucleoside mono- and diphosphate kinase can provide all four dNTPs since TK2 and DGUOK can only provide dNMPs.

Response: We have changed it according to the comment.

5. Figure legends should provide more information to make them easier to understand!

Response: We have provided more information into the figure legends.

6. Page 8 line 164, the authors should clarify “relative organ weights”, relative to what?

Response: We have rephrased ‘’The relative organ weights….’’ as ’’The relative organ weights (absolute organ weight vs total body weight)…..’’.

7. Page 8 line 167, what does the “net weights” mean? Is it total body weight? If so, “body weight” should be used instead.

Response: We have replaced “net weights” with “body weight”.

8. Page 10 line 215, “the products of TK2, dThd and dCyd, are”. dThd and dCyd are substrates of TK2 therefore this sentence need to rephrase.

Response: We have replaced “products” with “substrates”.

9. Page 10 line 218 and 219, “substitution” should be “supplementation”.

Response: We have replaced “substitution” with “supplementation”.

10. Page 10 line 226, “catabolism” should be “metabolism” since converting ribonucleotides to deoxyribonucleotides is an anabolic process”.

Response: This part has been deleted according to the comments from the reviewer 2.

11. Page 10 line 227 and 228, please provide experimental data/evidence or reference for how cells and organs share or transport nucleoside monophosphates.

Response: This part has been deleted according to the comments from the reviewer 2.

12. Page 10 line 232, “it seems like that the level kept is low but stable through the life span”. What are the mtDNA levels at earlier stage of life? Have the authors measured mtDNA levels at difference ages to support this statement?

Response: This part has been deleted according to the comments from the reviewer 2.

For the reviewer 2:

The major points from the reviewer 2:

1, I wish the authors could contextualize and compare the results presented in this manuscript metabolism to their previous data with the whole-body knockout. Could the authors add any more data on lipid metabolism in the reported mice that they can relate to the observed in the whole-body knockouts? Did the whole-body knockout present with a similar mtDNA depletion in the liver? The authors propose compensatory mechanisms, perhaps fuelling dNTP synthesis by TK2-independent pathways, even if from other tissues, that would explain the stabilization of mtDNA levels in liver-specific knockout mice. It would be very interesting to provide any data supporting that hypothesis. Have the authors measured deoxynucleoside levels in the liver or plasma? I think the authors should consider cytosolic phosphorylation of dCtd and dThd by deoxycytidine kinase (dCK) as a likely contributor to maintaining liver mtDNA levels in the tissue-specific knockouts.

Response: We have tried our best to contextualize our results from the livTk2 KO mice and compared them to our previous work from the total Tk2 knockout mice. In our total Tk2 knockout mice, the mtDNA level was around 20% of wild type mice at 14 days. However, in the livTk2 KO mice, the mtDNA levels were around 30% in male and 40% in female mice compared to wild type mice. We have added this information into the discussion part of the manuscript. We also performed experiments to address the reviewer’s questions which can be seen in the response to reviewer 1.

2, On the other hand, I miss the author’s discussion on other mouse models with a similar or even bigger degree of mtDNA depletion in the liver, some generated in their lab (i.e Dguok knockout mice) who have not shortened survival in spite of also manifesting with major alterations in lipid metabolism. Could demands on liver mitochondrial function or mtDNA copy number be different between human and mice? 

Response: In our Dexoguanosine kinase knockout (Dguok-/-) mouse model (Hum Mol Genet. 2019 Sep 1;28(17):2874-2884.), the mtDNA level can decline to around 5% of wild type mice at 8-weeks old. Compared to the livTk2 KO mice, these mice showed a clear body weight loss after 4-weeks old and an increased lipid catabolism. Similar to Dguok-/- mice, the livTk2 KO mice also showed a tendency of body weight loss. Therefore, these data suggested that mouse body weight paralleled with liver mtDNA level. Most human patients with DGUOK mutations died before 3 years old and Dguok-/- mice survived relatively longer. We believe that it might be due to species difference. There might be a different demand for liver mitochondrial function or mtDNA copy number between human and mice. 

3, The authors claim that ‘These mice were born normal and thus evidently had the ability to synthesize mtDNA during the fetal stage’. I wonder if this could have anything to do with the profile of the liver-specific cre-recombinase expression. Is it active from birth? Before? Have the authors checked for mtDNA copy number at any earlier timepoint?

Response: In the experiment, we used albumin-cre mice from Jackson lab to generate liver specific Tk2 knockout mouse model. It has been demonstrated that the albumin Cre transgene is effective in fetal and neonatal mice. Therefore, the TK2 activity in the livers in our livTk2 KO mice should be disrupted at birth. We have added the information into the discussion part. We did not check mtDNA copy number at any earlier timepoint. 

4, It is not clear why the authors choose to perform most of the analyses in liver and heart. Have they checked for the cre-recombinase efficiency in the liver? Perhaps by assessing TK2 protein or Tk2 RNA levels in the liver? Is it possible residual mtDNA (30%) comes at least partly, from not knocked-out hepatocytes/liver cells? Have they check for the cre-recombinase specificity? Perhaps checking TK2 protein/RNA expression in tissues other than liver, i.e heart and kidney?

Response: Based on our previous study in total Tk2-/- and Dguok-/- mice, we found that skeletal muscle and heart were equally affected by TK2 or DGUOK knockout. However, the variation in mtDNA copy number in skeletal muscle was very big compared to heart tissue with absolute real-time qPCR method. Therefore, we chose the heart as a control organ. 

We did not have a good antibody to TK2. We had tried several primary antibodies against TK2 in our previous study (PLoS One. 2022; 17(6): e0270418.), but we did not get a clear band for TK2 protein. In this study, we also tried to determine the Tk2 mRNA in liver tissues with qPCR. But it is difficult to design a pair of suitable primers since the exon 5 (which is deleted in the livTk2KO mice) is very short. The reverse primer for Tk2 gene in the study has 4 3’-bases located in exon 4, which might also explain why we can still detect Tk2 in livTk2KO mice. For the residual 30% mtDNA, the likelihood was very low that they came from hepatocytes since the albumin-cre transgene has been confirmed to be effective in fetal and neonatal ages.

Minor points:

1. In order to determine whether the statistical value has been correctly assessed sample size should be included for all the experiments. Also please specify whether the error bars indicate an error or standard deviation or better show individual samples distribution (for instance with a dot blot type of graph)

Response: We have included the sample sizes into figure legend of each individual figure. We also indicated that the error bars represent standard deviation in the relevant figures.

2, The authors mention a very limited number of knockout mice included in the weight monitoring graph. Could they explain why the number is so little if survival is not reduced?

Response: In total we included 8 livTk2 KO mice (n=4 for each gender) in the body weight monitoring. We tried to get more livTk2KO mice, but it took time to generate them, and we also did not have enough space in our animal facility. 

3, I find some observations to not be fully supported by the data:

* ‘The body weights of the male control mice were higher on average than in the livTk2 KO mice’. I understand from the graph that such a difference is not statistically validated.

Response: The reviewer is right about the body weight. It was an inadvertent mistake and we have rephrased the sentence from ‘The body weights of the male control mice were higher on average than in the livTk2 KO mice’ to ‘For the male mice, the average body weight of the livTk2 KO mice was not significantly changed compared to that in male control mice’.

*‘The investigated mitochondrial proteins, COX1 and SDH, showed similar levels in the livTk2 KO mice and the control mice, in both liver and heart tissue’. The authors show no quantification of the mentioned proteins but based on the included images, one can appreciate some differences between genotypes that are not discussed by the authors (i.e. increased heart COXI in the KO).

Response: The western blot was developed in machine Syngene G:BOX Chemi XX6, unfortunately, the machine was broken, and the stored original western blots results were lost, so we were not able to do the quantification with the software installed in the machine. However, even if there was a difference, the difference is not very evident as the results from liver tissues from our Dguok-/- mice (Hum Mol Genet. 2019 Sep 1;28(17):2874-2884.). To be on the safe side, we claimed that the livTk2 KO mice had similar levels of COX1 and SDH to the wild type mice.

4, Also, it could be advisable to check for the level of expression of additional respiratory chain subunits, even if nuclear-encoded, since steady-state levels will reflect synthesis but also stability and this may be variable between proteins. BN-PAGES or activity assays on full respiratory complexes would help generate a broader picture of the molecular phenotype in these mice.

Response: The suggestion is very nice; we will try to develop the BN-PAGES in our future study. Actually, we had also tested mtDNA-encoded ND1 peptide level in livTk2 KO mice, unfortunately, the ND1 antibody was not working. 

5, Males are known to weigh more than age-matched females: ‘The average of male mice body weights was higher than the female mice, when comparing control group male mice to control female mice, and livTk2 KO male mice to livTk2 KO female mice’. Thus, I suggest the authors reformulate this sentence (i.e. As expected, ….).

Response: We have removed the part in the manuscript.

6, Figure 1.

Could the authors locate in the diagram the position of the primers used in the genotyping in Fig 1D?

Response: We have labelled the position of the primers in Fig1D.

- Figure 4:

7, Can the images resolution be improved?

What do the scale bars indicate in units? Comparison of Wt and KO images should be performed on images captured at an equivalent scale preferably (I am not sure this is the case).

Also, the heart images may not be representative for the author’s conclusion, since they claim to observe no differences between WT and KO mice while the included images could suggest a different morphology for mitochondria in both genotypes). Could the authors comment on that? It could be helpful to add some arrows indicating the described features, especially in the HE stainings.

Response: We have re-done figure 4. We have chosen images with higher resolution and added proper scale bar units. For the heart electron microscope images, the difference is due to the longitudinal section in wild type and transverse section in livTk2 KO mice. We also added arrows to indicate the described features.

8, Line 204 ‘Total Tk2 knockout mice survived for three weeks’ and line 238 ‘Since the total Tk2 KO mice only survived approximately three weeks’.

It could perhaps be more accurate to talk about average survival and not maximal survival. I think on average, whole-body Tk2 knockout mice live around 16 days.

Response: The reviewer is correct about the live span of total TK2 knockout mice. As the reviewer said, the life span is around two weeks. The live span description in the manuscript was an inadvertent mistake. We have rephrased the sentences: Line 204 ‘Total Tk2 knockout mice survived for approximately two weeks’ and line 238 ‘Since the total Tk2 KO mice only survived approximately two weeks’.

9, I think in Line 209 the authors refer to cite number 7 in the reference list

Response: We have relisted the references accordingly.

We appreciate the constructive criticism from both Reviewer and are grateful for the opportunity to improve our manuscript. We hope that the revised manuscript is acceptable for publication in PlosOne. 

Yours sincerely 

Xiaoshan Zhou

---

## [Decision Letter · Decision Letter 1]

13 Aug 2023

Long term survival and abnormal liver fat accumulation in mice with specific thymidine kinase 2 deficiency in liver tissue

PONE-D-23-11585R1

Dear Dr. Zhou,

We’re pleased to inform you that your manuscript has been judged scientifically suitable for publication and will be formally accepted for publication once it meets all outstanding technical requirements.

Kind regards,

David C. Samuels

Academic Editor

PLOS ONE

Additional Editor Comments (optional):

Reviewers' comments:

Reviewer's Responses to Questions

**Comments to the Author**

1. If the authors have adequately addressed your comments raised in a previous round of review and you feel that this manuscript is now acceptable for publication, you may indicate that here to bypass the “Comments to the Author” section, enter your conflict of interest statement in the “Confidential to Editor” section, and submit your "Accept" recommendation.

Reviewer #1: All comments have been addressed

2. Is the manuscript technically sound, and do the data support the conclusions?

Reviewer #1: Yes

3. Has the statistical analysis been performed appropriately and rigorously? 

Reviewer #1: Yes

4. Have the authors made all data underlying the findings in their manuscript fully available?

Reviewer #1: Yes

5. Is the manuscript presented in an intelligible fashion and written in standard English?

Reviewer #1: Yes

6. Review Comments to the Author

Reviewer #1: In the revised manuscript the authors have addressed all my comments and I have no further comments.

7. PLOS authors have the option to publish the peer review history of their article (what does this mean?). If published, this will include your full peer review and any attached files.

Reviewer #1: No

---

## [Editor Report · Acceptance letter]

27 Sep 2023

PONE-D-23-11585R1 

Long term survival and abnormal liver fat accumulation in mice with specific thymidine kinase 2 deficiency in liver tissue 

Dear Dr. Zhou:

I'm pleased to inform you that your manuscript has been deemed suitable for publication in PLOS ONE. Congratulations! Your manuscript is now with our production department. 

Kind regards, 

on behalf of

Dr. David C. Samuels 

Academic Editor

PLOS ONE